# Attitudes toward Nursing Profession and Fear of Infectious Diseases among Undergraduate Nursing Students: A Cross-Sectional Study

**DOI:** 10.3390/healthcare11020229

**Published:** 2023-01-12

**Authors:** Abdulaziz Mofdy Almarwani, Fahad M. Alhowaymel, Naif S. Alzahrani, Hanan F. Alharbi

**Affiliations:** 1Department of Psychiatric Nursing, College of Nursing, Taibah University, Medina 42353, Saudi Arabia; 2Department of Nursing, College of Applied Medical Sciences, Shaqra University, Shaqra 11911, Saudi Arabia; 3Department of Medical-Surgical Nursing, College of Nursing, Taibah University, Medina 42353, Saudi Arabia; 4Department of Maternity and Child Health Nursing, College of Nursing, Princess Nourah bint Abdulrahman University, Riyadh 11564, Saudi Arabia

**Keywords:** fear, attitude, students, nursing, Monkeypox, COVID-19, Saudi Arabia

## Abstract

The uncertainty surrounding the effect of infectious diseases on nursing students’ attitudes toward the nursing profession worldwide exists. This study sought to examine the association between fear of infectious diseases and students’ attitudes toward the nursing profession. Methods: This quantitative descriptive cross-sectional study used a convenience sample of 477 students enrolled in nursing programs from three universities located in urban and non-urban areas in Saudi Arabia. Results: The study revealed a positive attitude toward nursing and minimal fear of infectious diseases. Junior nursing students reported significantly better attitudes and preferences toward the nursing profession than senior nursing students. Students who reported positive attitudes toward the nursing profession significantly had no intension to discontinue or transfer from nursing programs, whereas students with low attitudes reported significant intention to discontinue their enrollment in the nursing programs. Students in urban universities reported higher significant fear of infectious diseases compare to non-urban. The study revealed a significant negative correlation between fear of infectious diseases and students’ preference for the nursing profession. Conclusions: Nurse educators need to support senior nursing students’ attitudes toward the nursing profession and reduce their fear of infectious diseases, particularly among students studying in urban areas.

## 1. Introduction

Nursing is the largest healthcare profession worldwide [1,2]. Nurses’ roles are essential to the healthcare system, particularly with the increasing demand for qualified healthcare providers who can promote the quality of local and global healthcare services. The emergence of infectious diseases, such as COVID-19 and Monkeypox, have highlighted the importance of the nursing profession [3,4,5]. However, the widespread shortage of nurses is a global issue and requires immediate attention [6]. The International Council of Nursing (ICN) estimates that approximately 13 million nurses will be needed in the near future to respond to the aftermath of the COVID-19 pandemic [7]. Therefore, promoting enrollment and education of nursing students is essential [6]. Given the recent infectious disease outbreaks, it is important to understand students’ current attitudes toward nursing, and if fear of infectious diseases may influence their decisions to continue to pursue nursing as a career.

### 1.1. Attitudes toward the Nursing Profession

Attitudes toward the nursing profession are defined as students’ opinions about the nursing profession as a preference and priority in their future careers [8]. Understanding nursing students’ attitudes might help us better understand how future nurses perceive their profession and what attributes they value as necessary, appropriate, and relevant to nursing. A study conducted at public hospitals in Ethiopia to determine nurses’ attitudes toward their profession revealed that more than 50% of the participants reported negative attitudes [9]. In contrast, a study in Sarajevo reported positive attitudes toward the nursing profession, particularly for nursing students enrolled full-time compared with those for part-time students [10]. Similarly, a study was conducted in Turkey by Gol (2018) to examine anxiety and nursing students’ attitudes toward the profession, revealing positive attitudes [11]. One study assessing the level of community awareness and public image of the nursing profession in Saudi Arabia revealed low awareness and poor public image among Saudi caregivers [12]. This study was focused on public perceptions, but may have indicated that the public view may have influence over whether or not Saudi students decide to pursue nursing as a career. Limited research has been conducted on Saudi nursing students’ attitudes in light of the recent COVID-19 pandemic and other infectious diseases.

### 1.2. Nursing Fear of Infectious Diseases

Like most other healthcare professionals, nurses have been greatly impacted by the COVID-19 pandemic and the emergence of Monkeypox [13,14]. In Saudi Arabia, a study conducted to explore nurses’ fear during the COVID-19 outbreak revealed moderate levels of fear [15]. Many nurses reported a stronger fear of infectious diseases, especially after the pandemic [16]. There are several reasons for fear among nurses, including fear of being infected by diseases due to their direct work with patients, not being able to provide appropriate care for patients due to limited experience or resources, transmitting diseases to loved ones, and stigmatization and public image [17]. It is critical for nursing education systems to evaluate the current level of fear among nursing students, since this fear may impact their attitudes toward the nursing profession and could limit the desire for students to pursue nursing in the future. It is unclear whether nursing students’ attitudes toward their future profession have been affected by infectious diseases such as COVID-19 and Monkeypox.

### 1.3. Study Purpose

This study aimed to explore the level of attitudes toward the nursing profession and fear of infectious diseases. Additionally, this study sought to examine the association between fear of infectious diseases and students’ attitudes toward nursing.

## 2. Materials and Methods

### 2.1. Study Design

A quantitative descriptive cross-sectional design.

### 2.2. Study Sample and Settings

This study used convenience sampling of 477 undergraduate nursing students from three universities in Saudi Arabia. Data were collected from three different Saudi universities taking into consideration the geographical variations between the settings. Two universities are located in urban areas. One is in the capital city, Riyadh, located in the country’s central region. The other one is in Medina, located in the country’s western region. The third university is located in a suburban area in the central region of Saudi Arabia.

### 2.3. Research Instruments

The researchers collected demographic data, including sex, age, current bachelor’s level, university location, reason for becoming a nurse, and whether students had information before enrolling in the nursing program from different source such as their family, friends, social media, or university advertisements. We also asked one question to understand whether students thought of switching from nursing due to the COVID-19 pandemic and other infectious diseases.

The Attitude Scale for Nursing Profession (ASNP), developed in 2010 by Coban and Kasikci, was used to measure students’ attitudes toward the profession [8]. The attitude toward the nursing profession is defined as students perceived attitude toward their profession, including their perceived properties, preference toward working as nurses, and their position toward the profession [8]. The ASNP is comprised of 40 items with three subscales: (1) properties of the nursing profession (n = 18 items), (2) preference for the nursing profession (n = 13 items), and (3) general nursing profession position (n = 9 items) [8]. According to Coban and Kasikci, (2010), a mean score of three and above indicates a positive attitude and a mean score of less than three indicates a negative attitude. The instrument is considered reliable, with a Cronbach’s alpha of 0.9 [8].

The Fear of COVID-19 Scale (FCV-19S) was used to measure the fear of infectious diseases after some of its items were modified to focus on infectious diseases such as Monkeypox. The FCV-19S consists of seven items presented on a 5-point Likert scale. The instrument is considered reliable, with a Cronbach’s alpha of 0.82 [18]. 

### 2.4. Data Collection and Recruitment Strategies

The XXX University Institutional Review Board approved this study before data collection. An online survey was sent to students via email and other social media. The inclusion criterion was students who were currently enrolled in an undergraduate nursing program. Students not in a nursing program or those under 18 years of age were excluded.

### 2.5. Statistical Analysis

The data were analyzed using IBM SPSS Statistics version 25.0. Descriptive statistics including mean, standard deviation, and frequency were calculated. Independent sample *t*-tests, one-way ANOVA, and Pearson r correlation coefficients were performed to analyze the study data.

## 3. Results

### 3.1. Demographic Characteristics

This study included 477 undergraduate nursing students from various universities (Table 1), most of whom were women (73.8%) and in their third year of (mean age = 20.6 years, SD = 1.43). Most nursing students had information about nursing before their enrollment into the program (65.4%), and their family members were the primary sources of nursing information (42.1 %). Slightly more than half of the nursing students reported that they were either unsure or intended to discontinue their enrollment in the nursing program (51.1%).

### 3.2. Nursing Students’ Attitudes toward the Nursing Profession

Nursing students’ attitudes toward the nursing profession were measured using the ASNP, which has three subscales. The results revealed that nursing students had a positive attitude toward the nursing profession (M = 165, SD = 14.9). Nursing students reported their priority (M= 79.2, SD = 7.08), preference (M = 52.6, SD = 8.34), and general position in the nursing profession (M = 33.4, SD = 2.84) (Table 2).

### 3.3. Nursing Students’ Fear of Infectious Diseases

Nursing students reported their fear of infectious diseases using the FCV-19S, with scores ranging from 7 to 35. According to this scale, the higher the score, the stronger the fear of infectious diseases. The mean score for students’ fears of infectious diseases was 17.4 (Table 2).

### 3.4. Differences between Student Characteristics and Their Attitude toward the Nursing Profession and Fear of Infectious Diseases

The study revealed no statistical differences between the participants’ gender. One-way between-subject ANOVA was conducted on students’ attitudes toward the nursing profession, fear of infectious diseases, and academic years (first, second, third, fourth, and internship year). The test showed a significant difference in fear of infectious diseases with students’ academic years (F (4, 472) = 7.76, *p* < 0.001). Students in their third academic year reported a stronger fear of infectious diseases compared to other students in different academic years (M = 19.5, SD = 5.86). Moreover, students’ current academic year significantly affected their attitudes toward the nursing profession (F (4, 472) = 2.71, *p* = 0.002). Students in their first academic year showed more positive attitudes toward the profession than others (M = 169, SD = 16.1). Students’ current academic year had a significant effect on their preference for the nursing profession (F (4, 472) = 3.50, *p* = 0.008). Students in their first academic year showed a higher preference for the nursing profession than others (M = 55.0, SD = 9.22) (Table 3).

Additionally, students in university 2, which is in a large urban city of Saudi Arabia, reported significantly higher differences in fear of infectious diseases than other universities (M = 18.6, SD = 5.7, F (4, 472) = 6.08, *p* < 0.001). Students’ information about the nursing profession significantly affected their attitudes toward it (F (4, 472) = 8.4, *p* < 0.001). Students who reported having prior information about the nursing profession had more positive attitudes (M = 167, SD = 14), properties (M = 79.8, SD = 6.6, F (4, 472) = 3.97, *p* = 0.01), and preference for the profession (M = 53.8, SD = 8.1, F (4, 472) = 11.2, *p* < 0.001) (Table 3).

There was a significant difference between students’ intention to discontinue their enrollment in nursing programs and their attitudes toward the nursing profession (F (4, 472) = 12.9, *p* < 0.001). Students who did not want to discontinue their enrollment in the nursing programs had better attitudes toward the nursing profession (M = 168, SD = 14), followed by hesitant students who were undecided whether they wanted to discontinue their enrollment in the nursing programs (M = 166, SD = 14) and those who wished to discontinue their enrollment in the nursing program (M = 160, SD = 14) (Table 3).

Similarly, there was a significant difference between students’ intention to discontinue their enrollment in nursing programs and nursing profession preference (F (4, 472) = 32.0, *p* < 0.001). Students who did not want to discontinue their enrollment in nursing programs reported a higher preference for the nursing profession (M = 55.0, SD = 7.0), followed by hesitant students who were undecided whether they wanted to discontinue their enrollment in the nursing programs (M = 53.3, SD = 6.9) and students who wanted to discontinue their enrollment in the nursing program (M = 48.6, SD = 9.2) (Table 3).

Finally, an independent sample *t*-test was conducted to compare attitudes toward the nursing profession and fear of infectious diseases depending on the students’ university locations. There was a significant difference in fear of infectious diseases and students’ scores for students in urban (M = 17.8, SD = 5.9) and non-urban areas (M = 15.9, SD = 5.2; t (475) = 2.73, *p* = 0.007) (Table 3). There were no significant differences in the other variables. Pearson’s r correlation revealed a significant, negative correlation between fear of infectious diseases and students’ preference for the nursing profession (r (475) = −0.116, *p* = 0.011).

## 4. Discussion

This study evaluated undergraduate nursing students’ attitudes toward the nursing profession, their level of fear regarding infectious diseases, and how they may be related. Results showed that most students had a positive attitude toward the nursing profession, a moderate fear of infectious disease, and a negative association between students’ preferences for the nursing profession and fear of infectious diseases.

The majority of the study participants were female, in their third academic year, and received most of their information about nursing from their family members. As identified by many students, family members played an essential role in selecting nursing as a career. These results support previous findings, which suggest that parents can have influence over their children’s attitudes toward different professions and subsequently influence their career choice [19].

The study findings indicated that nursing students have positive attitudes toward nursing, similar to previous findings of another study conducted in Saudi Arabia using the ASNP [20]. Nursing students reported positive attitudes toward the nursing profession (M = 150). However, Miligi and Selim (2014) did not report the ASNP subscale scores, limiting the comparison between their results [20]. Another study in Turkey also found positive attitudes toward the nursing profession among students (M = 4.1 out of 5) [11]. The highest subscale scores were for students’ properties (M = 4.45), general position (M = 4.03), and preference for the nursing profession (M = 3.65). Studies in other countries measuring nursing students’ attitudes toward the nursing profession using different instruments have reported similar results to this study. For example, a study in Sarajevo used the Nursing Image Questionnaire to measure students’ opinions on the nursing profession’s duties, responsibilities, ethics, and social stereotypes [10]. Most students in this study had a positive attitude toward the profession. In contrast, the findings of our study are different from those found among Ethiopian nurses who reported mostly negative attitudes toward the nursing profession [9]. This could be due to organizational and cultural factors affecting their disposition [9].

Findings from this study indicated that junior nursing students have more positive attitudes toward the nursing profession than senior ones. In addition, the preference for nursing students entering the program was higher among junior students than senior ones. These findings contradicted those reported by Cukljek et al. (2017) [21] and Allari (2020) [22]. According to Cukljek et al. (2017), nursing students’ attitudes toward the profession altered over time, affecting the students regarding acquiring nursing knowledge and skills, including students’ progress during the nursing program [21]. Their attitudes toward the nursing profession became more positive over time. Additionally, Allari (2020) concluded that as nursing students progress in the program, they become more proud of their profession, indicating a positive attitude [22]. One possible explanation for these contradictory findings is the different circumstances nursing students currently face. Nursing students’ negative attitudes toward the profession as they progress in the program might be influenced by the emergence of the COVID-19 pandemic and related factors such as fear of infectious diseases [23]. Other factors may include online distance learning during the pandemic, limiting students from actively practicing the profession, and increasing uncertainty toward nursing [24].

Nursing students had a moderate level of fear of infectious diseases. This finding is similar to that of Moussa et al. (2021), who conducted a study to measure nurses’ fear of COVID-19 in Saudi Arabia using the FCV-19S. The study revealed a moderate level of COVID-19 fear (M = 19.7) [15]. The current study showed that students in their third academic year reported a stronger fear of infectious diseases than those in other academic years. One explanation for this result is that in Saudi Arabia, most nursing students actively practice in hospitals during their third academic year. In earlier academic years, nursing students practice in labs and simulation environments. Conducting this study during the COVID-19 pandemic likely contributed to the increased fear of infectious diseases. Moreover, nursing students in urban areas reported a stronger fear of infectious diseases than students studying in non-urban areas. This may be because urban areas were more affected by COVID-19 than non-urban areas [25].

Fear of infectious diseases was found to be negatively affecting nursing students’ preference for the nursing profession. This is consistent with prior research, which has shown that nursing students with greater fear of diseases such as HIV/AIDS, Ebola, and COVID-19 have negative attitudes toward nursing [26,27,28].

Future research is needed to examine factors influencing fear of infectious diseases and nursing students’ preference for the profession and develop effective interventions to promote positive attitudes toward the nursing profession. For example, one study in India developed an intensive four-day health education program on HIV/AIDS, which significantly improved nurses’ knowledge, reduced fear, and improved their attitudes [29]. Given the recent surges in infectious diseases such as COVID-19 and Monkeypox, the implementation of programs similar to this one may be effective in improving students’ attitudes toward nursing.

### 4.1. Implications for Nursing Education

Factors influencing the nursing education process should be examined regularly. Influencing variables that might affect students’ attitudes toward their future careers, such as their increasing fear of infectious diseases, need to be considered [15]. Nurse educators have several responsibilities during students’ enrollment in the nursing program, such as guiding and helping students prepare for nursing practice and becoming successful nurses [30]. They are also responsible for assessing students’ educational needs [31] and other factors that might influence their attitudes toward their future profession [9]. Therefore, this study provides evidence regarding the relationship between students’ attitudes toward the nursing profession and their fear of infectious diseases. This study can help nurse educators build future knowledge regarding students’ attitudes and fear of infectious diseases. Based on this study’s findings, nurse educators should be aware of senior students’ fear of infectious diseases, which might affect their continuity in the nursing program and work as registered nurses (RN) in the future. Nurse educators might consider building special educational programs before students enroll in their first clinical training. These programs should aim to improve students’ understanding of the precautions used to prevent infectious diseases. Moreover, the programs should focus on reducing students’ fear of infectious diseases and improving overall attitudes toward nursing.

### 4.2. Study Limitations

While this study yielded useful information, it also has some limitations. The use of a cross-sectional design limits the ability to determine causal relationships. Additionally, bias could emerge due to the use of convenience sampling. Finally, this study was specific to nursing students at three universities in Saudi Arabia, which limits the generalizability of the results to a broader population.

## 5. Conclusions

Attitude toward the nursing profession is an essential contributor influencing nursing students’ decisions to pursue nursing as a career. An additional factor influencing students’ attitudes is the fear of infectious diseases due to the COVID-19 pandemic and the emergence of its variants and other contagious diseases, such as Monkeypox [13]. Nursing students showed positive attitudes toward the nursing profession, with junior students showing more positive attitudes and preferences than senior students. Students who reported having prior information about the nursing profession had a positive attitude, properties, and preferences compared to students who did not have any information about the nursing profession. Most students who had no intention of discontinuing their enrollment in nursing programs had a higher positive attitude toward the profession. Finally, nursing students in this study had a moderate fear of infectious diseases, with third-year students having the most fear compared to other students. Future research is needed to identify ways to mitigate fear of infectious diseases among undergraduate nursing students and examine other factors that could play a key role in student attitudes toward the profession.

## Figures and Tables

**Table 1 healthcare-11-00229-t001:** Demographic characteristics (N = 477).

Measure	N	M	SD
Age (years)	477	20.6	1.43
**Measure**	**N**	**%**	
Sex			
Male	125	26.2	
Female	352	73.8	
Current Nursing Year			
First Year	67	14.0	
Second Year	140	29.4	
Third Year	155	32.5	
Fourth Year	77	16.1	
Internship Year	38	8.0	
University			
University 1	152	31.9	
University 2	235	49.3	
University 3	86	18.0	
Other	4	0.80	
University Locations			
Urban	387	81.1	
Non-Urban	90	18.9	
Information about Nursing Prior Enrollment			
Yes	312	65.4	
No	118	24.7	
Not Sure	47	9.9	
Source of Nursing Information			
Family Members	201	42.1	
Social Media	128	19.3	
University	92	26.8	
Friends	56	11.7	
Thinking about Changing Nursing Major			
Yes	158	33.1	
No	233	48.8	
Maybe	86	18.0	

**Table 2 healthcare-11-00229-t002:** Level of attitude toward nursing profession and fear of infectious diseases (N = 477).

Measure	M	SD	Range
Fear of Infectious Diseases (7 items, Cronbach’s α = 0.87)	17.4	5.89	7–35
Attitude toward Nursing Profession (40 items, Cronbach’s α = 0.79)	165	14.9	40–200
Properties of the Nursing Profession Subscale (18 items, Cronbach’s α = 0.82)	79.2	7.08	18–90
Preference to Nursing Profession Subscale (13 items, Cronbach’s α = 0.86)	52.6	8.34	13–65
General Position of Nursing Profession Subscale (9 items, Cronbach’s α = 0.72)	33.4	2.84	9–45

**Table 3 healthcare-11-00229-t003:** Analysis of variance and independent sample t-test of selective students’ characteristics on mean difference ASNP and FCV-19S (N = 477).

Measure	N	FCV-19S			ASNP			Properties			Preference			General Position		
		M (SD)	t/F	*p*	M (SD)	t/F	*p*	M (SD)	t/F	*p*	M (SD)	t/F	*p*	M (SD)	t/F	*p*
Sex			0.70	0.48		0.04	0.96		0.42	0.67		0.79	0.42		1.05	0.29
Male	125	17.1 (5.6)			165.3 (15.3)			79.0 (7.4)			53.1 (7.9)			33.2 (2.9)		
Female	352	17.6 (5.9)			165.2 (14.8)			79.3 (6.9)			52.3 (8.4)			33.5 (2.8)		
Current Nursing Year			7.76	<0.001 ***		2.71	0.02 *		4.97	0.36		3.50	0.08		1.09	0.36
First Year	67	15.6 (6.0)			169 (16.1)			80.5 (7.1)			55.0 (9.2)			33.7 (2.9)		
Second Year	140	16.6 (5.1)			165 (15.9)			79.2 (7.9)			53.1 (8.0)			33.2 (2.8)		
Third Year	155	19.5 (5.8)			163 (14.3)			78.7 (6.8)			51.6 (7.8)			33.2 (2.9)		
Fourth Year	77	16.7 (5.6)			166 (13.4)			79.6 (6.4)			52.9 (7.9)			33.9 (2.6)		
Internship Year	38	17.1 (7.0)			160 (13.1)			77.9 (5.5)			49.4 (9.1)			33.2 (2.2)		
University			6.08	<0.001 ***		0.47	0.70		0.16	0.91		1.6	0.17		1.3	0.26
University 1	152	16.6 (6.1)			165 (15)			79.5 (6.4)			52.5 (9.4)			33.7 (2.7)		
University 2	235	18.6 (5.7)			165 (14)			79.1(6.9)			52.8 (7.6)			33.2 (2.9)		
University 3	86	15.9 (5.2)			164 (16)			78.4 (8.4)			52.8 (8.1)			33.5 (2.7)		
Other	4	16.0 (7.3)			157 (14)			79.7 (7.8)			43.5.(7.1)			34.0 (1.4)		
University Locations			2.73	0.007 **		0.52	0.60		0.44	0.65		0.60	0.54		0.74	0.73
Urban	387	17.8 (5.9)			165 (14.6)			79.3 (6.7)			52.7 (8.3)			33.4 (2.8)		
Non-Urban	90	15.9 (5.2)			164 (16.5)			78.8 (8.4)			52.1 (8.3)			33.5 (2.7)		
Information about Nursing Prior Enrollment			1.44	0.23		8.4	<0.001 ***		3.97	0.01 **		11.2	<0.001 ***		0.89	0.40
Yes	312	17.1 (5.8)			167 (14)			79.8 (6.6)			53.8 (8.1)			33.5 (2.8)		
No	118	18.2 (6.3)			162 (15)			78.3 (7.7)			50.3 (8.4)			33.5 (2.7)		
Not Sure	47	17.6 (4.7)			159 (15)			77.2 (7.9)			49.7 (7.3)			32.9 (3.3)		
Source of Nursing Information			0.40	0.74		2.04	0.10		1.91	0.12		1.93	0.12		0.08	0.96
Family Members	201	17.7 (5.7)			166 (14)			79.7 (6.9)			52.9 (7.9)			33.4 (2.7)		
Social Media	128	17.2 (5.6)			166 (14)			79.6 (6.7)			53.5 (8.0)			33.4 (2.9)		
University	92	17.0 (6.2)			163 (15)			78.4 (6.9)			51.6 (9.3)			33.5 (2.8)		
Friends	56	17.8 (6.5)			161 (16)			77.6 (8.2)			50.8 (8.5)			33.3 (3.1)		
Thinking about Changing Nursing Major			0.91	0.39		12.9	<0.001 ***		0.98	0.37		32.0	<0.001 ***		2.35	0.09
Yes	158	17.5 (6.4)			160 (14)			78.6 (6.6)			48.6 (9.2)			33.2 (2.5)		
No	233	17.2 (5.6)			168 (14)			79.6 (7.1)			55.0 (7.0)			33.3 (3.0)		
Maybe	86	18.2 (5.6)			166 (14)			79.2 (7.7)			53.3 (6.9)			34.0 (2.7)		

* = *p* ≤ 0.05; ** = *p* ≤ 0.01; *** = *p* ≤ 0.001. **FCV-19S**: The Fear of COVID-19 Scale. **ASNP**: Attitude Scale for Nursing Profession.

## Data Availability

The datasets used and/or analyzed during the current study are available from the corresponding author on reasonable request.

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
