# Peer review of "Attitudes toward Nursing Profession and Fear of Infectious Diseases among Undergraduate Nursing Students: A Cross-Sectional Study"

_healthcare, 2023, doi:10.3390/healthcare11020229_

Round 1
Reviewer 1 Report
This manuscript highlights an aspect of nursing education that needs more understanding. The impact of infectious disease on the attitudes of students towards the profession of nursing requires attention. I have made a few comments that will strengthen the ability of the manuscript to communicate effectively to the audience. This is especially important regarding some of the key concepts.

Author Response
Reviewer 1
Dear reviewer, we thank you for your time reviewing our manuscript. Your valuable and insightful comments led to possible improvements in the current version. The authors have carefully considered the comments and tried our best to address every one of them.
Please see below for point-by-point responses to the reviewers' comments.
Sincerely,
Authors.

Reviewer 2 Report
Thank you for the opportunity to review this work. The study was well described, logically organized. Please address the following issues:
On line 18. Abstract: “This qualitative descriptive explorative” should be changed to This quantitative…
Line 82 section 2.1. Study design: is there a reference for the STROBE check-
List used in this study?
Under discussion section, please address the implications for nursing education based on current results in this research study.
Author Response
Reviewer 2
We would like to thank you for your time reviewing our manuscript. Your valuable and insightful comments led to possible improvements in the current version. The authors have carefully considered the comments and tried our best to address every one of them.
Please see below for point-by-point responses to the reviewers' comments.
Sincerely,
Authors.

Reviewer 3 Report
Recommendations for the manuscript author(s) include:
A revision to the results table to clearly illustrate the study findings to the reader in relation to the participants’ gender (male, female).
For reader clarity, provide specific descriptions of the study findings for each measure in relation to gender (male, female) under the results section.
Revision of the typographical errors under the abstract are necessary.
For reader clarity, provide a description of the study participants’ setting (e.g. college, university, technical) and student level of nursing educational program (e.g. vocational, undergraduate, graduate).
There are minor grammar revisions within the body of the manuscript requiring attention and revisions.
Author Response
Reviewer 3
We would like to thank you for your time reviewing our manuscript. Your valuable and insightful comments led to possible improvements in the current version. The authors have carefully considered the comments and tried our best to address every one of them.
Please see below for point-by-point responses to the reviewers' comments.
Sincerely,
Authors.
